# Role of SIRT3 in Microgravity Response: A New Player in Muscle Tissue Recovery

**DOI:** 10.3390/cells12050691

**Published:** 2023-02-22

**Authors:** Michele Aventaggiato, Federica Barreca, Laura Vitiello, Simone Vespa, Sergio Valente, Dante Rotili, Antonello Mai, Lavinia Vittoria Lotti, Luigi Sansone, Matteo A. Russo, Mariano Bizzarri, Elisabetta Ferretti, Marco Tafani

**Affiliations:** 1Department of Experimental Medicine, Sapienza University of Rome, 00161 Rome, Italy; 2Laboratory of Flow Cytometry, IRCCS San Raffaele Roma, Via di Val Cannuta 247, 00166 Rome, Italy; 3Center for Advanced Studies and Technology, University “G. d’Annunzio” of Chieti-Pescara, Via Luigi Polacchi 11, 66100 Chieti, Italy; 4Department of Drug Chemistry and Technologies, Sapienza University of Rome, 00185 Rome, Italy; 5MEBIC Consortium, San Raffaele University, 00166 Rome, Italy; 6Cellular and Molecular Pathology, IRCCS San Raffaele Roma, Via di Val Cannuta 247, 00166 Rome, Italy

**Keywords:** simulated microgravity, sirtuins, SIRT3, muscle tissue, ROS, mitochondria, molecular rehabilitation

## Abstract

Life on Earth has evolved in the presence of a gravity constraint. Any change in the value of such a constraint has important physiological effects. Gravity reduction (microgravity) alters the performance of muscle, bone and, immune systems among others. Therefore, countermeasures to limit such deleterious effects of microgravity are needed considering future Lunar and Martian missions. Our study aims to demonstrate that the activation of mitochondrial Sirtuin 3 (SIRT3) can be exploited to reduce muscle damage and to maintain muscle differentiation following microgravity exposure. To this effect, we used a RCCS machine to simulate microgravity on ground on a muscle and cardiac cell line. During microgravity, cells were treated with a newly synthesized SIRT3 activator, called MC2791 and vitality, differentiation, ROS and, autophagy/mitophagy were measured. Our results indicate that SIRT3 activation reduces microgravity-induced cell death while maintaining the expression of muscle cell differentiation markers. In conclusion, our study demonstrates that SIRT3 activation could represent a targeted molecular strategy to reduce muscle tissue damage caused by microgravity.

## 1. Introduction

Microgravity is a singular condition in which normal gravity (1 g) is altered [1]. This situation occurs, for example, in space or in the International Space Station (ISS). On ground, two main devices can be used to reproduce the effects of weightlessness [2]. Random Positioning Machine (RPM) and Rotary Cell Culture System (RCCS) simulate microgravity in different way. RPM rotates around two axes creating a situation in which the gravity vector can be approximated to zero [3,4,5]. RCCS is a vertically rotating machine in which cellular spheroids and cellular aggregates rotate simulating microgravity conditions and the “free fall” that occurs in space [6]. During spaceflight, microgravity causes several physiological effects on post mitotic tissues like cardiovascular and respiratory system, bone, muscle, immune system, brain etc. enhancing cancer risk [1,7,8,9,10,11]. Postural muscles, in human and animal are affected by microgravity maintaining only 70% of their initial mass after 270 days in simulated or real microgravity [12]. Furthermore, several studies have demonstrated that microgravity promotes the degradation of muscle structural proteins and an inhibition of stem cells differentiation [13,14] suggesting the presence of a direct effect of microgravity in skeletal muscle atrophy. The most important change involves the slow twitch fibers that mutate in the fast type [12,15]. Microgravity causes also “deconditioning” [16], that is characterized by loss in force and power of muscle and abnormal reflex patterns due to an abnormal size of fibers without altering their number. Human gastrocnemius and soleus lose most of their mass while other muscles such as tibialis and knee extensors lose only a low percentage of their mass [12]. This is not an irreversible condition. After reloading, in fact, it is possible to reproduce the correct homeostasis of the organisms following a personal exercise schedule and with Advanced Resistive Exercise Device (ARED) [17,18,19,20]. At the muscular level, the damage involves Z-bands and the endomysium with microcirculatory rearrangement and interstitial edema [12]. Myogenic differentiation is regulated by several master genes that, once activated, cause the commitment of cells to the myogenic lineage. They are named basic helix-loop-helix myogenic regulatory transcription factors (MRFs) and include MyoD, Myf5, myogenin and MRF4 [21,22,23]. Biochemical signals play an important role in myogenesis regulation, however under microgravity, MRFs expression is altered [24]. Furukawa et al. demonstrated that muscle differentiation is strictly connected to the expression of MRFs factors and reported a role for DNA methylation as a key factor in gravity-regulating myogenesis [24]. On the other hand, several works also demonstrated the important role of ROS signaling for fate decisions in adult stem cells. Notably, a pro-oxidative setting increases myocyte differentiation [25]. In long-term quiescent satellite cells, differentiation process is inhibited by low ROS and is favored by high ROS level [26,27]. N-acetyl-cysteine (NAC) or other antioxidants inhibit ROS creating a reductive environment preventing muscle differentiation [28]. However, the interplay between ROS and the molecular pathways that are involved in gravity sensing are still largely unknown [24]. Taken together these data suggest that a more detailed investigation is necessary to clarify the role of microgravity in the differentiation of muscle tissue. An important candidate in this complex scenario, may be represented by Sirtuins, a class of NAD+-dependent deacylases and mono-ADP-ribosyltransferase enzymes [29] activated by calorie restriction, natural compounds (resveratrol, honokiol etc.) and exercise [30] that represent a fundamental bridge between metabolism and epigenetic regulation [31]. These enzymes revealed a strong homology with Saccharomyces Cerevisiae class of protein called Sir2 and involved in lifespan increase [32]. Seven sirtuins have been identified in mammals, with a different localization and function: SIRT1 and SIRT6 are nuclear sirtuins, SIRT2 is cytoplasmatic, SIRT3, SIRT4 and SIRT5 are mitochondrial and finally SIRT7 is nucleolar [32]. Among the three mitochondrial sirtuins, SIRT3 controls key metabolic pathways because it is involved in the respiratory chain, ATP production, citric acid and urea cycles, and, insulin secretion, which are very important for muscle differentiation, stability and adaptation [33]. Recent studies have shown that mitochondria influence muscle development because are strictly connected with myoblast proliferation and differentiation and with the acquisition, in physiological conditions, of the contractile and metabolic features of muscle fibers [34,35,36]. Abdel Khalek et al. in fact, demonstrated that SIRT3 increases during differentiation of myotubes until the formation of muscle fibers. They also demonstrated that the same situation involves the increased expression of Myogenin, MyoD and PGC1α until the last day of differentiation [37]. On the contrary, SIRT1 shows a decreasing pattern during the differentiation process [36]. In addition, SIRT3 depletion causes the inhibition of C2C12 terminal differentiation [37]. Therefore, mammalian mitochondrial sirtuins are emerging enzymes connected with cellular metabolism since they represent metabolic sensors that modulate the activity of several targets via deacylation or mono-ADP-ribosylation [38]. In this way, compounds that can increase sirtuins’ enzymatic activity, represent a new frontier to modulate cellular metabolism and cellular differentiation. In this study we detailed the inductive effect of microgravity on vitality, differentiation, stemness and autophagy in C2C12 cells and, partially on the more sensitive H9C2 cells, using a RCCS system (Rotary Cell Culture System) to reproduce microgravity on ground. Moreover, using a new SIRT3 synthetic activator, we investigated the interplay between microgravity and SIRT3 functions in muscle tissue and the possibility to revert the negative effects of microgravity on this tissue.

## 2. Materials and Methods

### 2.1. RCCS (Rotary Cell Culture System)

To simulate microgravity conditions, a RCCS machine has been used. This device is characterized by a continuous rotation of the vessels that simulate microgravity on the principle that continuous rotation constantly changes cell orientation with respect to gravity reproducing the free fall occurring in Space or on International Space Station (ISS). RCCS is a rotating machine with bubble free vessels and a membrane for diffusion gas exchange and characterized by the absence of shear stress among spheroids and cellular aggregates and the culture medium. The clinostat used in these experiments was purchased by Synthecon Incorporated (Houston, TX, USA) and provides a vector-averaged reduction on gravity force acting on cells of 10^−3^ g when vessels rotate at 10 rpm compared with normogravity conditions (1 g) [39]. Once differentiated, C2C12 or H9C2 cells (1 × 10^6^) were injected into a 10 mL chamber and all air bubbles were carefully removed to prevent fluid shear stress. Subsequently, the effect of microgravity was achieved by rotation around the horizontal axis at 10 rpm for 24 and 48 h according to the treatments protocol described in this section and using as control the same conditions not subjected to rotation.

### 2.2. Cell Culture

Mus musculus skeletal muscle cells (C2C12) were purchased from LGC Standards (Milan, Italy) and were grown in Dulbecco’s Modified Eagle’s Medium with high glucose (D6546, Sigma-Aldrich–MERCK, St. Louis, MO, USA) supplemented with 10% Fetal Bovine Serum (F9665, Sigma-Aldrich–MERCK, St. Louis, MO, USA), 2 mMol/L-glutamine (G7513 Sigma-Aldrich–MERCK, St. Louis, MO, USA), 100 units/mL penicillin and 0.1 mg/mL streptomycin (P0781 Sigma-Aldrich –MERCK, St. Louis, MO, USA). Adherent cells were detached by Trypsin-EDTA solution (T4049, Sigma-Aldrich–MERCK, St. Louis, MO, USA). Cell line was maintained at 37 °C in a humidified atmosphere of 5% CO_2_ and 95% air. Differentiation into myotubes was achieved by culturing C2C12 myoblasts in complete medium and then switching into DMEM medium supplemented with 2% FBS for 72 h when cells were 90% confluent. H9C2 rat cardiomyoblast cells were purchased from ATCC and were grown in Dulbecco’s Modified Eagle’s Medium with high glucose. This culture medium was supplemented with 10% Fetal Bovine Serum, 2 mMol/L-glutamine, 100 units/mL penicillin and 0.1 mg/mL streptomycin. Adherent cells were detached by Trypsin-EDTA solution. H9C2 cells were maintained at 37 °C in a humidified atmosphere of 5% CO_2_ and 95% air. H9C2 cells were then differentiated in low-serum (1% FBS), retinoic acid (RA)-supplemented (10 nM) DMEM for 7 days beginning 24 h after seeding [40]. RA (R4643 Sigma-Aldrich–MERCK, St. Louis, MO, USA) was dissolved in DMSO according to manufacturer’s instructions and stored at −20 °C. Differentiation medium was replaced every other day while fresh RA was added every day to the medium and the unused portion discarded.

### 2.3. Treatments Protocols

SIRT3 activator (MC2791), synthesized and validated as previously described [41], was dissolved in dimethyl sulfoxide (DMSO; D2438, Sigma-Aldrich–MERCK, St. Louis, MO, USA) and added at the final concentration of 50 μMol/L every 8 h for 24 h and 48 h. The same concentration of DMSO was used as control. N-Acetyl-L-Cysteine (A9165-5G; Sigma-Aldrich–MERCK, St. Louis, MO, USA), was added at the final concentration of 1 mMol/L [42,43] for 24 and 48 h.

### 2.4. Protein Extraction and Immunoblotting

Cells for whole cell lysate were centrifuged at 555× *g* for 10 min at 4 °C and pellets were resuspended in a solution containing 50 mMol/L Tris-Cl (Tris-Cl; 93352, Sigma-Aldrich–MERCK, St. Louis, MO, USA), 250 mMol/L sodium chloride (NaCl; S7653, Sigma-Aldrich–MERCK, St. Louis, MO, USA), 0.1% Triton^®^ X-100, 5 mMol/L ethylenediaminetetraacetic acid (EDTA; E6758, Sigma-Aldrich –MERCK, St. Louis, MO, USA) and 0.1 mMol/L Dithiothreitol (DTT; D9163, Sigma-Aldrich–MERCK, St. Louis, MO, USA) plus 1 mMol/L phenylmethylsulfonyl fluoride (PMSF; 93482, Sigma-Aldrich–MERCK, St. Louis, MO, USA), 1 mMol/L sodium orthovanadate (Na_3_VO_4_; S6508, Sigma-Aldrich–MERCK, St. Louis, MO, USA) and 10 mMol/L sodium fluoride (NaF; 201154, Sigma-Aldrich–MERCK, St. Louis, MO, USA), Protease inhibitor cocktail (PI; P8340 Sigma-Aldrich–MERCK, St. Louis, MO, USA). After 30 min on ice, samples were centrifuged at 14,000× *g* for 10 min at 4 °C and the supernatants collected. Protein concentration was determined by Bradford assay (500-0205, Bio-Rad, Hercules, CA, USA). Clarified protein lysates (50 μg) were boiled for 5 min, electrophoresed onto denaturating SDS-PAGE gel and transferred onto a 0.45 µm nitrocellulose membrane (162-0115, Bio-Rad, Hercules, CA, USA). The blotting membranes were blocked with 5% nonfat dry milk (1706404, Bio-Rad, Hercules, CA, USA) for 1 h at room temperature (rt) and then incubated with primary antibody overnight at 4 °C. The following day, membranes were washed three times with 0.1% Tween^®^ 20 in PBS (PBST) for 30 min rt and incubated with the appropriate secondary antibody for 1 h rt. After another three washes in PBST, the detection of the relevant protein was assessed by enhanced chemiluminescence (Lite Ablot^®^ TURBO; EMP012001 EuroClone, Milan, Italy). Densitometric analysis of the bands, relative to β-Actin, was determined by ImageJ Software (National Institutes of Health). The following primary antibodies were used for Western Blot analysis: LC3 (NB600-1384, Novus Biologicals, Centennial, CO, USA), β-ACTIN (A5316, Sigma-Aldrich–MERCK, St. Louis, MO, USA), SIRT1 (PA5-17074, Thermo Scientific, Waltham, MA, USA), MyHβ (A4840) (sc-166930 Santa Cruz, Dallas, TX, USA), MyHα (K13) (sc-168676 Santa Cruz, Dallas, TX, USA), MHC (sc-20641 Santa Cruz, Dallas, TX, USA), Pax-7 (sc-81648 Santa Cruz, Dallas, TX, USA), SIRT3 (D22A3) (5490S Cell Signaling, Danvers, MA, USA), BNIP3 (sc-56167, Santa Cruz, Dallas, TX, USA), YAP (sc-376830, Santa Cruz, Dallas, TX, USA). Horseradish peroxidase-linked anti-mouse (NA931V) and anti-rabbit (NA934V) antibodies were purchased from GE Healthcare.

### 2.5. Measurements of Reactive Oxygen Species

Cells were seeded onto RCCS vessels and incubated in microgravity conditions for 24 and 48 h with SIRT3 activator added every 8 h and with the other treatments described in this section. The same protocol was used for cells seeded into 100 mm dishes in normogravity conditions. After 24 and 48 h of treatment in normogravity and microgravity, cells were incubated with MitoSOX Red (Mitochondrial superoxide indicator, Invitrogen, Life Technologies Corporation, Eugene, OR, USA) dissolved in DMSO at the final concentration of 5 µMol/L for 10 min, according to manufacturer’s instructions. Subsequently cells were collected and washed twice in HBSS/Ca/Mg (Gibco 14025-092, Budapest, Hungary). Fluorescence was read by CytoFlex flow cytometer (Beckman/Coulter) and Median Fluorescence Intensity (MFI) has been considered for graphic analysis. ROS production in H9C2 was evaluated using Glomax^®^-Multi Detection System (Promega, Madison, WI, USA) and fluorescence intensity at 625/660-720 nm was graphed as shown in Appendix A as a result of three different sets of experiments.

### 2.6. Propidium Iodide Staining

To evaluate cell death, cells were analyzed by flow cytometry. Cells were seeded in normogravity conditions into 100 mm dishes and in microgravity conditions into RCCS vessels and treated as described above. Cells were subsequently harvested with trypsin-EDTA, washed twice with ice cold PBS, centrifuged at 800× *g* for 5 min at 4 °C and, finally fixed with pre-cold 70% ethanol overnight at 4 °C. Samples were stained with 50 μg/mL Propidium Iodide (PI, P4864; Sigma-Aldrich–MERCK, St. Louis, MO, USA) in PBS for 2 h at 4 °C cover light. Fluorescence was read by BD FACSDiva 8.0.2 flow cytometer (Becton Dickinson, Milan, Italy). The sub-G1 fraction, which represents the total amount of apoptotic cells, was determined and analyzed through CellQuest™ software.

### 2.7. Viability Assays

Cell viability after normogravity and microgravity treatments was assessed with Clonogenic Assay and Trypan Blue Exclusion Assay. In the colony assay, at the end of treatments, cells (5 × 10^2^) were plated in 100 mm dishes in culture medium with 10% FBS to perform the clonogenic assay. At the end of treatments, plates were washed twice with a phosphate buffered saline solution (PBS; 79382; Sigma-Aldrich–MERCK, St. Louis, MO, USA) and fixed with 4% formaldehyde solution in PBS (F8775; Sigma-Aldrich–MERCK, St. Louis, MO, USA) at rt. After 20 min, dishes were washed twice in PBS and stained for 5 min with 0.5% crystal violet (C0775; Sigma-Aldrich–MERCK, St. Louis, MO, USA). Finally, cells were washed with distilled water and air-dried. The colonies were counted the following day. In trypan blue exclusion assay, cells were harvested and stained with 0.4% trypan blue (T8154; Sigma-Aldrich–MERCK, St. Louis, MO, USA). The cell suspension was applied to a haemocytometer and counted with a phase contrast microscopy (NIKON Eclipse TE2000U, Nikon Netherlands, Amsterdam, The Netherlands).

### 2.8. Immunofluorescence and Fusion Index Calculation

C2C12 cells (5 × 10^4^) were seeded in 10% FBS onto coverslip placed inside 35 mm dishes. The day after cells were placed in 2% FBS conditions for 48 h during which they were either left untreated or treated with MC2791, NAC and NAC+MC2791. Alternatively, differentiated cells were first exposed to 24 h of microgravity in the presence or absence of indicated treatments and then counted and seeded onto coverslip placed inside 35 mm dishes in differentiating medium in normogravity for 48 h. Subsequently, cells were fixed with 4% formaldehyde solution in PBS for 10 min, washed three times with PBS and permeabilized for 5 min in 0.1 M Tris pH 7.4 and 0.2% Triton^®^ X-100. After three washes in PBS, samples were incubated for 1 h at rt with 0.2 mg/mL BSA followed by 2 h at rt with anti-MHC antibody at 1:500 dilution in BSA (sc-20641 Santa Cruz, Dallas, TX, USA). Cells were then washed three times with PBS and incubated for 1 h with the secondary antibody goat anti-rabbit IgG Alexa Fluor^®^ 555 at 1:200 dilution in BSA (A21429; Invitrogen, Carlsbad, CA, USA). Samples were washed again for three times in PBS and nuclei were labeled with SYTO™ green (ThermoFisher Scientific, Munich, Germany) at 1:20,000 dilution in PBS. Finally, after three washes in PBS, samples were mounted using ProLong^®^ Diamond Antifade Mountant (P36961; ThermoFisher Life Technologies, Waltham, MA, USA) and analyzed with a LSM510 confocal microscopy (Zeiss, Oberkochen, Germany). Fusion Index was measured manually according to Hinkle et al. [44].

### 2.9. Optical Microscope Analysis

Using Zeiss IM35 microscope (Zeiss) and a digital camera (Nikon Digital Sight DSL1), we determined the dimensions of the cells in control conditions after treatment with DMSO, MC2791, NAC and NAC plus MC2791. Dimensions are expressed in μm.

### 2.10. Electron Microscopy

C2C12 cells were cultured in 10 mm dishes in normogravity conditions and in RCCS vessels in microgravity conditions and treated as described for 48 h. Cells were washed with warm PBS and fixed with 2.5% glutaraldehyde solution (G7651; Sigma-Aldrich–MERCK, St. Louis, MO, USA) in 0.1 M sodium cacodylate buffer pH 7.3 (C0250; Sigma-Aldrich) at 4 °C overnight. The following day, samples were collected, washed three times with cacodylate buffer and fixed for 2 h rt with 2% osmium tetroxide (75632; Sigma-Aldrich–MERCK, St. Louis, MO, USA) using the same buffer. Subsequently, samples were dehydrated with two washes of 10 min each in several solutions of different ethanol concentrations (50%, 70%, 90%, 100%) and finally embedded in Epon resin. Ultrathin sections obtained through a Reighert-Jung Ultra cut E ultramicrotome (Leica Microsystems, Wetzlar, Germany) were stained with uranyl-acetate replacement stain (EMS #22405; EMS, Hatfield, PA, USA) and lead hydroxide. A Jeol-1400-plus TEM was used for observation and photographic analysis.

### 2.11. Statistical Analysis

Data are presented as the mean of standard deviation, determined from three or more experiments per condition. Differences between pairs of groups were analyzed by Student’s *t*-test. The level of significance was set as * *p* < 0.05, ** *p* <0.01 and *** *p* < 0.001.

## 3. Results

### 3.1. Microgravity Increases Cell Death

Microgravity influences cell differentiation and proliferation increasing the apoptotic cell rate and negatively affecting the regenerative muscle growth [45]. To test the effects of microgravity on skeletal muscle cells we treated differentiated C2C12 in normal gravity (normogravity, 1 g) as well as in microgravity conditions (microgravity, 10^−3^ g). Compared to normogravity, control cells in microgravity show a higher percentage of cell death (Figure 1) as underlined by propidium iodide staining in sub-G1 phase after 48 h. The gating strategy adopted to exclude cellular debris and to select the right cell population is shown in Appendix A. Recently, we have validated a series of SIRT3 and SIRT5 specific activators [41]. To understand a possible involvement of the mitochondrial sirtuin 3 (SIRT3) in the molecular mechanisms that characterize response of post mitotic cells to microgravity, we analyzed, among these new selective SIRT3 activators, the effects of MC2791 (cpd 11 in [41]). Our choice is motivated by the fact that SIRT3 is tightly connected with myoblast proliferation and differentiation [46]. In fact, we observed that when differentiated C2C12 were kept in microgravity there was an increase of SIRT1 after 72 h and a decrease of SIRT3 at 4 and 72 h (Appendix A). In normogravity, sub-G1 analysis of cell cycle after 24 or 48 h of MC2791, NAC or NAC plus MC2791 treatment did not show any difference in the percentage of dead cells (Figure 1). We used NAC, throughout the study, to investigate the role of oxidative stress in normogravity and microgravity. Interestingly, when cells were kept for 48 h in microgravity, there was a significant increase in cell death (Figure 1). However, MC2791 treatment reduced the extent of such cell death compared to the control or to the other treatments (Figure 1).

The effects of SIRT3 activation by MC2791 in normogravity and microgravity were also studied in the rat cardiomyoblast cell line H9C2. Results shown in Appendix A evidence a decrease in cell death of differentiated H9C2 after 48 h treatment with MC2791 compared to the control. On the contrary, there was an increase of cell death after 24 and 48 h treatment with NAC and a corresponding decrease of cells in the G0/G1 and S phase of the cell cycle (Appendix A). Interestingly, differentiated H9C2 cells were extremely sensitive to microgravity to a point that almost 80% of untreated cell were dead after 24 and 48 h (Appendix A). Notwithstanding this, MC2791 was still capable of significantly reduce cell death after 48 h of microgravity in differentiated H9C2 cells (Appendix A). Interestingly, such protective effect was lost in the presence of NAC where the percentage of cell death was higher than control cells (Appendix A). Unfortunately, with such an effect of microgravity on H9C2 we were able to measure the production of ROS but not to perform clonogenic and immunoblotting assays.

### 3.2. SIRT3 Activation Prevents Microgravity Effects on Post Mitotic Cell Lines

Microgravity increases cell death in postmitotic myocytes and cardiomyocytes an event that is reduced by SIRT3 activation. Clonogenic assays were performed on differentiated C2C12 cells either left untreated or treated with MC2791, NAC or NAC+MC2791 in normogravity or microgravity for 24 and 48 h. Compared to control untreated cells, MC2791 increased the number of colonies after 24 and 48 h in normogravity. On the contrary, in simulated microgravity a decrease in colony number after MC2791 treatment was observed after 24 and 48 h (Figure 2). Treatments with NAC and NAC+MC2791 determined an increase of colony number after 24 h in microgravity (Figure 2). Interestingly, when studying the morphology of the cells within the colonies, we observed that MC2791 treatment caused an elongation of C2C12 cells suggesting the rapid re-activation of a differentiation process. In fact, increase in cell length was observed only in the presence of MC2791 under both 24 and 48 h of normogravity and microgravity (Figure 2). On the contrary, NAC alone reduced cell length after 48 h of microgravity (Figure 2).

Moreover, we also observed that MC2791 treatment accelerates C2C12 differentiation in normogravity as shown in Figure 3 where cells were plated in differentiating medium in the presence of MC2791, NAC or NAC plus MC2791 for 4 (T0), 24 (T1) and 48 h (T2). In particular, after 48 h, cells treated with MC2791 showed an elongated morphology. On the contrary, 48 h of NAC treatment reduced C2C12 differentiation (Figure 3). Differentiation of C2C12 cells after 48 h in normogravity was also confirmed by measuring fusion index after nuclei and MHC staining. Appendix A shows myotube assembly and fusion index of C2C12 differentiated cells. It is clear that MC2791 treatment increased the number of nuclei inside a myotube, the size of the myotube and, the fusion index compared to the other treatments. On the contrary, we observed a reduction of myotube formation and fusion index following NAC treatment (Appendix A). Therefore, we aimed to study the behavior of C2C12 cells once returned to normogravity after a period in microgravity. We did this in order to mimic a re-adaptation to normogravity. Interestingly, cells treated for 24 h in microgravity with MC2791 and with NAC+MC2791 were significantly elongated after 4 h of normogravity (T0) when compared with control cells and NAC treated cells (Figure 3). After 24 and 48 h in normogravity, cells treated with MC2791 showed a significant increase in size when compared with control cells and with other treatments. However, such a differentiating effect by MC2791 on C2C12 cells was lost after returning to normogravity following 48 h of simulated microgravity exposure, probably due to the prolonged microgravity exposure time (Figure 3). Also, in this case we determined the myotube morphology and measured fusion index. Appendix A shows how 24 h of microgravity followed by 48 h of normogravity reduced myotube formation as well as fusion index. However, MC2791 was still capable to maintain the myotube structure as well as to increase fusion index compared to the other treatments (Appendix A). We repeated the experiments on the H9C2 cell line. Due to the extensive cell death in microgravity, we did not obtain colonies in the clonogenic assay. However, we were able to study the effect of MC2791 treatment on H9C2 cells in normogravity. Interestingly, in H9C2 cells, MC2791 treatment, either alone or plus NAC, decreased the number of colonies after 48 h (Appendix A). Similar to C2C12 cells, morphological examination of the H9C2 cells in the colonies, revealed an increase in H9C2 cell size after 24 and 48 h of MC2791 and after 24 h of MC2791 plus NAC treatment (Appendix A). Moreover, MC2791 treatment also increased intracellular fibers content after 48 h (Appendix A white arrows).

### 3.3. SIRT3 Activation Effects on Skeletal Muscle Cells Differentiation in Normogravity and Microgravity Conditions

Based on our previous results, we developed the hypothesis that modulation of SIRT3 activity could be used to reduce the effects of microgravity on muscle cells. In order to test our hypothesis, C2C12 differentiated cells were kept in normogravity and microgravity for the times indicated and the expression of stemness and differentiation markers determined. In particular, we evaluated expression of Myosin Heavy Chain (MHC), MyHα, MyHβ, YAP, and Pax-7.

The MyHβ/MyHα ratio can be used to determine the level of differentiation of muscle cells with a higher ratio indicating a higher stemness state [47]. Our results show that MC2791 treatment decreased MyHβ/MyHα ratio after 24 and 48 h of microgravity (Figure 4A). On the contrary, MyHβ/MyHα ratio increased when cells were treated with NAC after 24 and 48 h of microgravity (Figure 4A). Finally, NAC plus MC2791 treatment reduced MyHβ/MyHα ratio after 48 h of microgravity (Figure 4A). MHC is considered a marker of late differentiation of myoblasts [48]. Again, compared to control or NAC treated cells, MC2791 treatment increased MHC expression after both 48 h of normogravity or microgravity (Figure 4B).

Furthermore, we also determined the expression of YAP, a transcriptional co-activator considered a stemness factor blocking the differentiation of skeletal muscle cells [49]. As shown in Figure 4C, YAP expression in normogravity decreased after 24 h of MC2791 and NAC+MC2791 treatment compared to control cells. Such a trend was evident also after 48 h of treatment with MC2791 in normogravity. Interestingly, after 24 and 48 h of microgravity exposure, MC2791 treatment decreased YAP expression if compared to control cells (Figure 4C). Again, NAC treatment increased YAP expression, especially after 48 h of microgravity. Along with YAP, we also measured the expression of the transcription factor Pax-7 considered a specific factor stimulating proliferation and inhibiting differentiation of muscle satellite cells [50]. Our results show that microgravity increased Pax-7 expression in C2C12 cells after 24 and 48 h (Figure 4D). Importantly, MC2791 significantly reduced Pax-7 expression under microgravity compared to control or NAC treated cells (Figure 4D).

### 3.4. SIRT3 Activation Effects on ROS in Normogravity and Microgravity

It is well known that SIRT3 is responsible of ROS reduction with protective role against oxidative stress-dependent disorders [51,52]. We analyzed ROS production in normogravity and microgravity conditions in the presence of MC2791. We first compared ROS production in proliferating versus differentiated C2C12 cells treated or untreated with MC2791 in normogravity conditions (Figure 5A). In proliferating C2C12 cells, MC2791 treatment stimulates ROS production both at 24 and 48 h (Figure 5A). ROS production in differentiated control C2C12 cells is higher than in proliferating control cells (Figure 5A). Figure 5B shows the effect of MC2791 treatment on differentiated C2C12 cells in normogravity and microgravity conditions. Moreover, we also used NAC as antioxidant control. After 24 and 48 h in normogravity, MC2791 intensified ROS production if compared with control cells, while NAC reduced oxidative stress [41]. Interestingly, NAC and MC2791 cotreatment increased ROS production. In microgravity, we observed a ROS increase after 24 h and 48 h in the presence of MC2791 (Figure 5B). NAC treatment had no effect in preventing ROS increase both at 24 and 48 h of microgravity (Figure 5B). Moreover, at 48 h of microgravity we observed a ROS increase with NAC treatment (Figure 5B). Interestingly, after 48 h of simulated microgravity exposure, altered ROS production highlights a double peak in control and treated cells probably due to the presence of two cellular populations (Figure 5B), a situation previously reported in other cell lines [53]. In order to clarify the ROS increase observed with the SIRT3 activator MC2791, we treated C2C12 cells with the SIRT3 inhibitor 3-TYP [54]. As shown in Figure 5C, 3-TYP did not increase ROS levels in differentiated C2C12 after 24 and 48 h of normogravity (Figure 5C). In the case of H9C2 cells, in normogravity, MC2791 treatment resulted in an initial decrease of ROS after 24 h followed by an increase after 48 h compared to control cells (Appendix A). NAC treatment reduced ROS both after 24 and 48 h (Appendix A). Similar results were observed in microgravity with two clarifications: (i) the ROS signal was very low (100 times lower than normogravity) due to the extensive cell death as documented in Appendix A; (ii) the ROS accumulation in the presence of MC2791 was present after both 24 and 48 h of treatment (Appendix A).

### 3.5. SIRT3 Activation Increases Autophagy and Decreases Mitophagy in Normogravity and Microgravity Conditions

Several studies demonstrate that autophagy plays an important role in myoblast differentiation [55]. In fact, autophagy inhibition results in cell death and both autophagy and mitophagy are required for myoblast differentiation [56,57]. As shown in Figure 6A, SIRT3 activation through MC2791 treatment increased autophagy marker LC3II both at 24 and 48 h in normogravity conditions compared to control cells. NAC treatment, instead, decreased LC3II expression while NAC and MC2791 cotreatment leaded to an increase of LC3II expression. In microgravity, MC2791 increased LC3II after 48 h (Figure 6A). On the contrary, NAC treatment decreased LC3II expression after 48 h of microgravity (Figure 6A). We also investigated the mitophagy process, through which damaged or dysfunctional mitochondria are removed, measuring the expression of BNIP3 a well-known mitochondrial cargo receptor [58,59]. As shown in Figure 6B, BNIP3 expression decreased in normogravity conditions after 24 and 48 h of MC2791 treatment. Such a BNIP3 decrease was lost after 48 h of NAC treatment in normogravity (Figure 6B). In microgravity MC2791 decreased BNIP3 expression after 24 h when compared to control cells (Figure 6B). Conversely, NAC and NAC plus MC2791 treatment increased BNIP3 expression. The same trend was seen after 48 h of microgravity exposure (Figure 6B).

Ultrastructural analysis of C2C12 cells under microgravity revealed an increased number and electron density of mitochondria after MC2791 treatment (Figure 7) suggesting a decrease in mitophagy as well as the presence of autophagosomes (Figure 7 lower panel). Moreover, only in the case of NAC and NAC plus MC2791 treatment we also observed the presence of mitophagic bodies containing or close to mitochondria (arrows and lower panel of Figure 7) suggesting the possible activation of mitophagy [60].

## 4. Discussion

In the present study, we propose SIRT3 activation as a potential mechanism to revert and limit the damage to skeletal muscle tissue due to microgravity exposure. As it is known, microgravity causes extensive damage to muscle tissue, bone tissue, immune system, circulation system, etc. In this complex scenario, our purpose was to identify a possible molecular mechanism that acts to interfere with microgravity-induced muscle cells damage, characterized by myocytes dedifferentiation, and able to restore cell homeostasis. Several works have shown that cells exposed to microgravity display changes in morphology, biochemical pathways and gene expression related not only to the gravity sensing but also to altered cytoskeleton architecture and unbalance with the microenvironment [61]. Furthermore, it is well known that microgravity causes an inhibition of cell differentiation and promotes a progressive dedifferentiation in several cell lineages [62]. These results underline the importance of understanding the cellular mechanisms that, once activated, may counteract dedifferentiation favoring the maintenance of a differentiated state and cellular functions. In this perspective, sirtuins represent a family of proteins of considerable importance for their wide-ranging activity on various cellular functions in physiological and pathological conditions [63]. In muscle tissue, mitochondrial sirtuin SIRT3 has a central role in mitochondrial metabolism regulation, cell survival, oxidative stress resistance and differentiation state maintenance [33]. Using MC2791, a new synthetic SIRT3 activator, we investigated the role of this sirtuin in microgravity conditions to revert cell damage and dedifferentiation in muscle tissue induced by weightlessness. In our experiments, we reported a protective effect of MC2791 in counteracting cell death connected to microgravity exposure. In fact, cell cycle analysis in microgravity showed a decrease in cell death after 48 h of MC2791 treatment (Figure 1). Interestingly, NAC treatment was detrimental and not protective against microgravity (Figure 1) notwithstanding the ability of NAC to reduce ROS levels as reported in several studies underlining the role of NAC in cell death regulation through different mechanisms such as inhibition of cell proliferation, G1 growth arrest, reduction in O2•− and H_2_O_2_ induced damage, etc. [64]. Microgravity also reduced colony formation both in the presence or absence of the SIRT3 activator MC2791. However, morphology analysis of C2C12 cells revealed two important aspects: (1) MC2791 treatment increased the length of the myocytes in the colonies (Figure 2) and (2) MC2791 treatment also increased the length and the fusion index of myocytes kept under microgravity and then replated in normogravity under differentiating conditions (Figure 3). Therefore, we can conclude that SIRT3 activation with MC2791 maintains the differentiation state of myocytes under microgravity and accelerates differentiation of myocytes returning to normogravity after being exposed to microgravity. While it is known that SIRT3 overexpression or activation is beneficial for terminally differentiated cells (cardiomyocytes, myocytes, neurons, etc.) [65,66,67], to our knowledge this is the first study to analyze the effect of SIRT3 activation in myocytes under microgravity both in term of cell survival and differentiation. Importantly, molecular analysis of differentiation and stemness markers, confirmed morphological observations, documenting a decrease in MyHβ/MyHα ratio and an increase in MHC after MC2791 treatment during 24 and 48 h of microgravity exposure and during 48 h of treatment in normogravity conditions (Figure 4A,B). Furthermore, we also documented a decreased expression of the stemness marker YAP after 24 and 48 h of MC2791 treatment both in normogravity and microgravity and a reduction in Pax-7 expression both at 24 and 48 h of MC2791 treatment in microgravity underlining a more differentiated state in muscle cells (Figure 4C,D). To further confirm our results, we analyzed ROS production in normogravity and microgravity conditions. Commonly, the drastic increase in ROS level is associated with mitochondrial damage and disruption and consequently with the induction of the apoptotic process. On the other hand, an increase of physiological ROS level is strictly associated with inhibition of cell self-renewal potential and an increase in differentiation state [68]. Myoblast differentiation is characterized by a significant level of cellular stress including elevated levels of ROS. In fact, repressing mitochondrial ROS production through antioxidant agents, inhibits the differentiation process [69]. Our results fit with these observations. In fact, we observed that MC2791 increased mitochondrial ROS production in both proliferating and differentiated C2C12 cells (Figure 5). Moreover, MC2791 treatment also increased mitochondrial ROS of differentiated cells both at 24 and 48 h in normogravity and microgravity conditions (Figure 5). Interestingly, ROS production was reduced by NAC in normogravity whereas it was increased in microgravity (Figure 5B,C). Although more experiments are required to unravel this peculiar result with NAC, we speculate that, in microgravity, the effect of NAC is lost or reversed, i.e., the same concentration of NAC can, in microgravity, increase instead of inhibit ROS production. Such ROS increasing effect has been reported, for example, when using high dosages of antioxidants and, in our case, could be due to the combination of NAC and microgravity. Another important observation was that after 48 h of microgravity we documented the presence of two seemingly separated cellular populations differing in the amount of ROS produced. In fact, in all the treatments, we observed a population with low and a population with high ROS content (Figure 5B). This agrees with what recently reported by Po et al., that described how the removal of gravity constraint sets free the cells to choose between a round floating state or an elongated adherent state [53]. Our observations suggest that these two populations might also differ in ROS content. We also investigated the effect of SIRT3 activation, through MC2791 treatment, on autophagy and mitophagy. Autophagy is, in fact, a highly preserved catabolic pathway involved in organelles and protein aggregates degradation to produce nutrients and drive cellular architecture changes thereby regulating metabolic stress adaptation, cell differentiation, diseases regulation etc. [70]. Although initially it was thought that the activation of the autophagic process was limited to the initial stages of muscle tissue differentiation, new studies have confirmed that autophagy characterizes the entire differentiation process, accompanied by an increase of LC3II expression, conversion from glycolytic to oxidative metabolism and an increase of mitochondria number [71,72,73]. In the present study we demonstrate that MC2791 increases LC3II expression at 24 and 48 h in normogravity conditions and at 48 h of microgravity exposure suggesting a role of SIRT3 activation in recycling intracellular material for the maintenance of a more differentiated state of muscle cell in response to microgravity (Figure 6). These data are confirmed by mitophagy analysis. In the present study we analyzed BNIP3 expression as a mitophagy marker revealing that MC2791 treatment both at 24 and 48 h in normogravity and after 24 h of microgravity exposure, caused an inhibition of mitophagy compatible with the maintenance of the mitochondrial population and function as well as of a more differentiated state of muscle cells (Figure 6). On the contrary, NAC treatment caused an increase in mitophagy process connected with a less differentiated state of muscle cells (Figure 6). Our results agree with what was demonstrated by Sin et al., i.e., that mitochondria are targeted for degradation in the early stage of muscle cells differentiation process and this process is dependent upon an intact autophagic flux [74,75]. Autophagy and mitophagy activation after MC2791 treatment was also confirmed through Transmission Electron Microscopy (Figure 7). TEM analysis also showed an increase in number and electron density of the mitochondria in MC2791 treated cells compared to control or NAC treated cells. Taken together our results demonstrate that SIRT3 activation could be considered a mechanism that, once activated, leads to a maintenance of differentiated state in muscle cells in order to counteract muscle atrophy or muscle tissue damage after weightlessness exposure. This represents an interesting discovery since, at the moment, the measures adopted to restore the physiologic conditions after spaceflights are based on ARED and personalized physical exercise [19,20]. Therefore, a targeted molecular intervention, in a perspective of damage recovery, could represent a more effective and faster solution that can be implemented during and after spaceflight.

## Figures and Tables

**Figure 1 cells-12-00691-f001:**
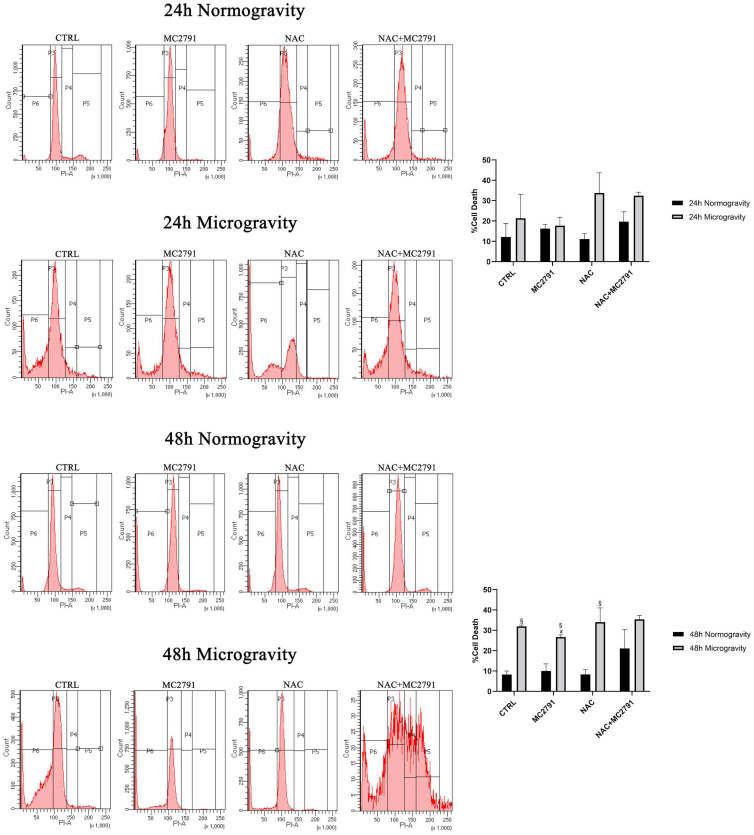
SIRT3 activation decreases cell death rate in C2C12 differentiated cells after 48 h of microgravity exposure. C2C12 cells were differentiated and subsequently treated with MC2791, NAC and NAC+MC2791 for 24 and 48 h in normogravity and microgravity conditions. The figure shows differences in the percentage of sub-G1 cells determined through propidium iodide staining and cytofluorimetric analysis. In this figure, P3 represents G0/G1 phase, P4 S phase, P5 G2/M phase and P6 represents the subG1 cells. Experiments were repeated three times. Differences between pairs of groups were analyzed by Student’s *t*-test. # Significantly decreased compared with untreated cells. #, *p* < 0.05. § Significantly increased compared with the same treatment in normogravity conditions. §, *p* < 0.05.

**Figure 2 cells-12-00691-f002:**
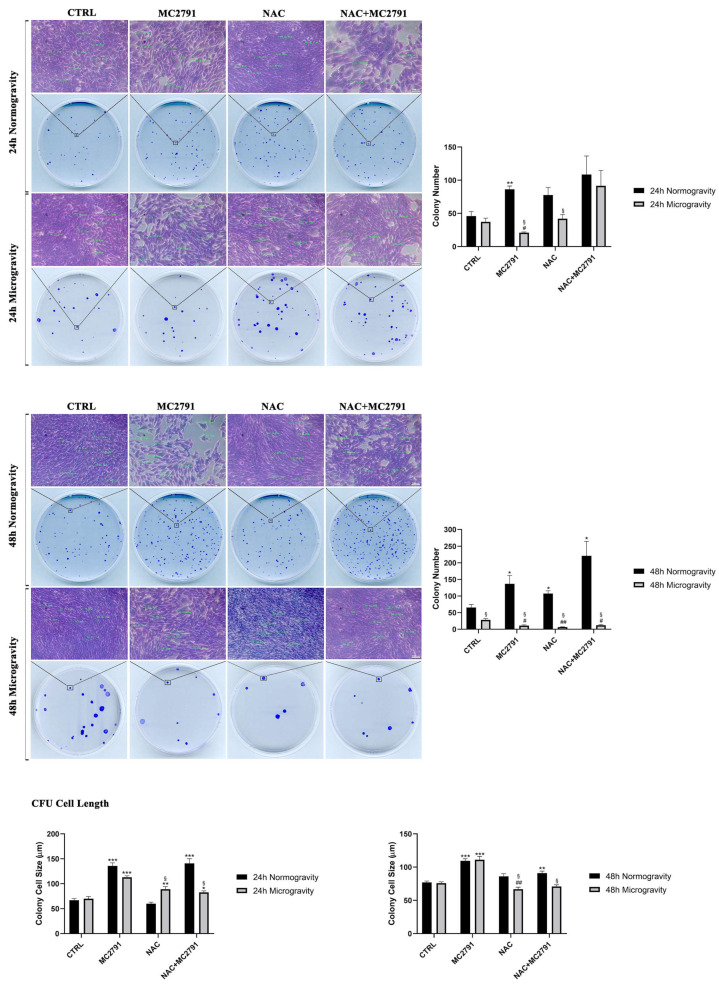
SIRT3 activation protects differentiated myocytes from microgravity related negative effects. Differentiated C2C12 cells were treated as described in the figure. Colonies were analyzed through a phase contrast microscopy at 20× magnification. Upper panel, colonies formation after treatment with MC2791, NAC and NAC+MC2791 compared with untreated cells after 24 h in normogravity and microgravity conditions. On the right side of upper panel, colonies count was graphed. Middle panel, comparison of the same treatments after 48 h of normogravity and microgravity exposure. On the right side of the middle panel, colonies number was graphed. Lower panel, analysis of cell morphology within colonies as reported in both graphs. Size bars are on the right bottom image of each panel. Data are representative of three separate experiments. Differences between pairs of groups were analyzed by Student’s *t*-test. * Significantly increased compared with untreated cells. *, *p* < 0.05. **, *p* < 0.01. ***, *p* < 0.001. # Significantly decreased compared with untreated cells. #, *p* < 0.05. ##, *p* < 0.01. § Significantly different in comparison with the same treatment in normogravity conditions. §, *p* < 0.05.

**Figure 3 cells-12-00691-f003:**
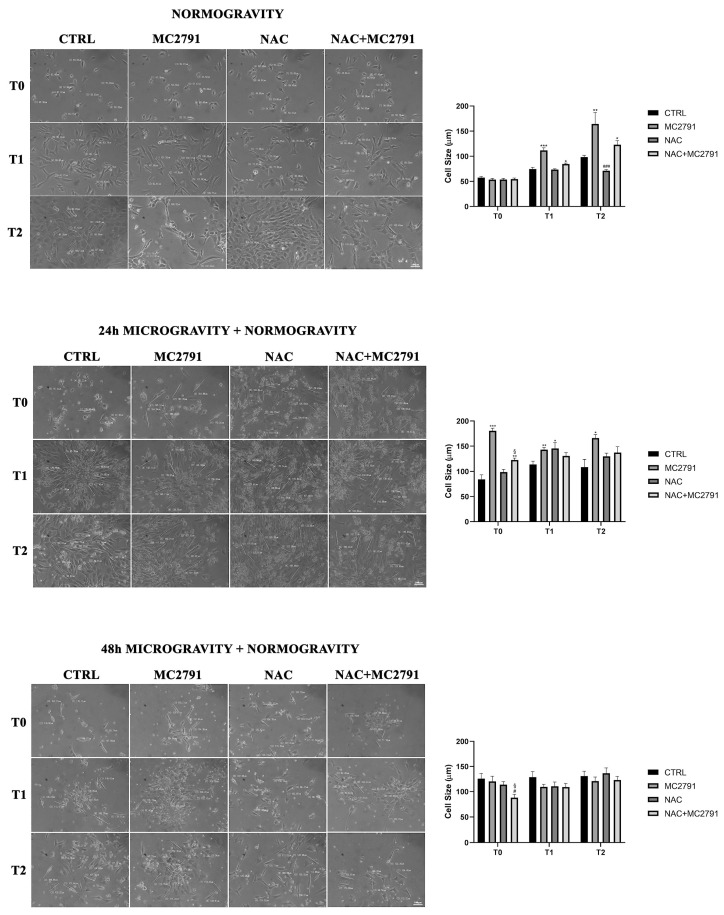
MC2791 promotes cell differentiation in normogravity conditions and the re-adaptation after microgravity exposure. In the upper panel, cells were grown in differentiating medium and treated as indicated. Cell size measurements were performed after 4 (T0), 24 (T1) and 48 (T2) hours of treatment. 20× magnification was used in order to underline morphology differences among treatments. In the middle panel, cells were replated in normogravity after 24 h of microgravity exposure and cell size measurements were performed after 4 (T0), 24 (T1) and 48 (T2) hours from plating at 20× magnification. In the lower panel, cells were replated in normogravity after 48 h of microgravity exposure and cell size measurements were performed after 4 (T0), 24 (T1) and 48 (T2) hours from plating at 20× magnification. Size bars are on the right bottom image of each panel. On the right side of each panel, cell size measurements were graphed. Data are representative of three separate experiments. Differences between pairs of groups were analyzed by Student’s *t*-test. * Significantly increased compared with untreated cells. *, *p* < 0.05. **, *p* < 0.01. ***, *p* < 0.001. # Significantly decreased compared with untreated cells. #, *p* < 0.05. ###, *p* < 0.001. § Significantly different in comparison with NAC treatment at the same time point. §, *p* < 0.05.

**Figure 4 cells-12-00691-f004:**
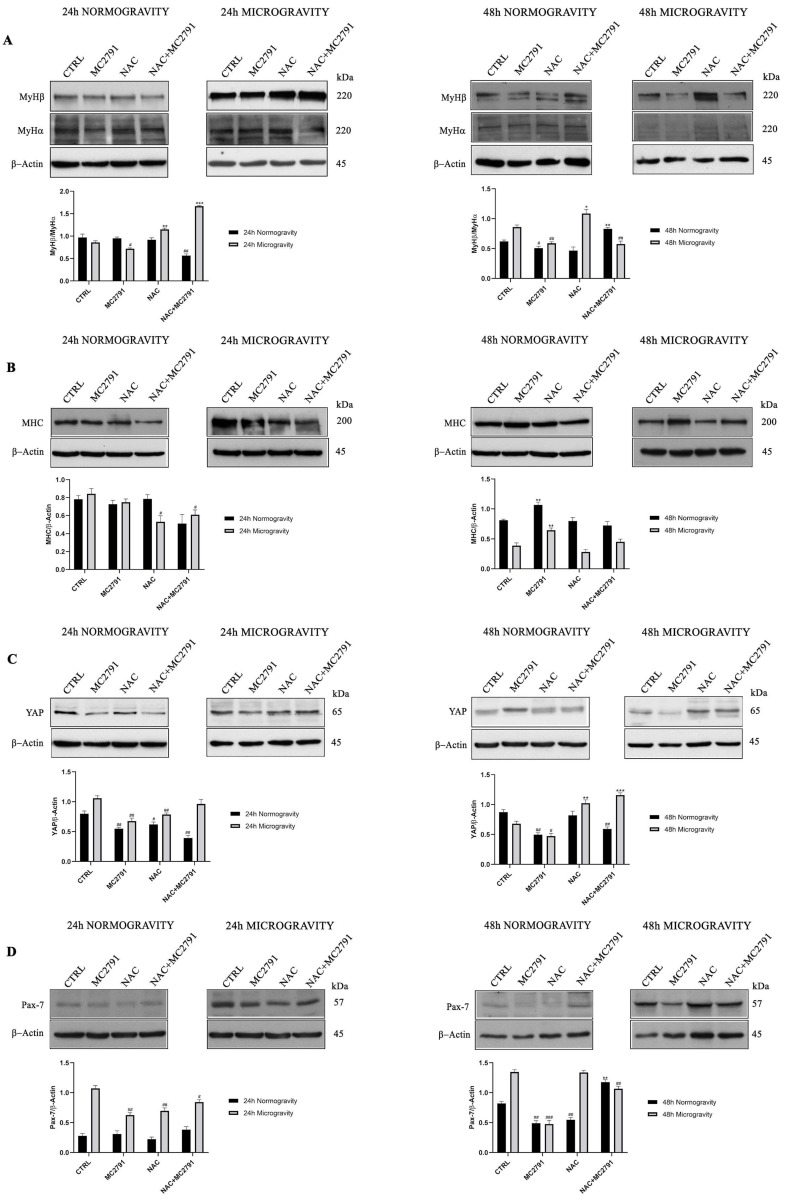
SIRT3 activation reduces microgravity effects on differentiated muscle cells. Cells were treated as indicated in the figure in normogravity and microgravity conditions (**A**,**B**) Expression levels of differentiation markers (MyHα, MyHβ and MHC) and (**C**,**D**) stemness markers (YAP and Pax-7) were determined by Western blot. β-Actin was used as a loading control. Densitometric analysis of the gels was performed as described under Materials and Methods. Data are representative of three separate experiments. Differences between pairs of groups were analyzed by Student’s *t*-test. * Significantly increased compared with untreated cells. *, *p* < 0.05. **, *p* < 0.01. ***, *p* < 0.001 # Significantly decreased compared with untreated cells. #, *p* < 0.05. ##, *p* < 0.01. ###, *p* < 0.001.

**Figure 5 cells-12-00691-f005:**
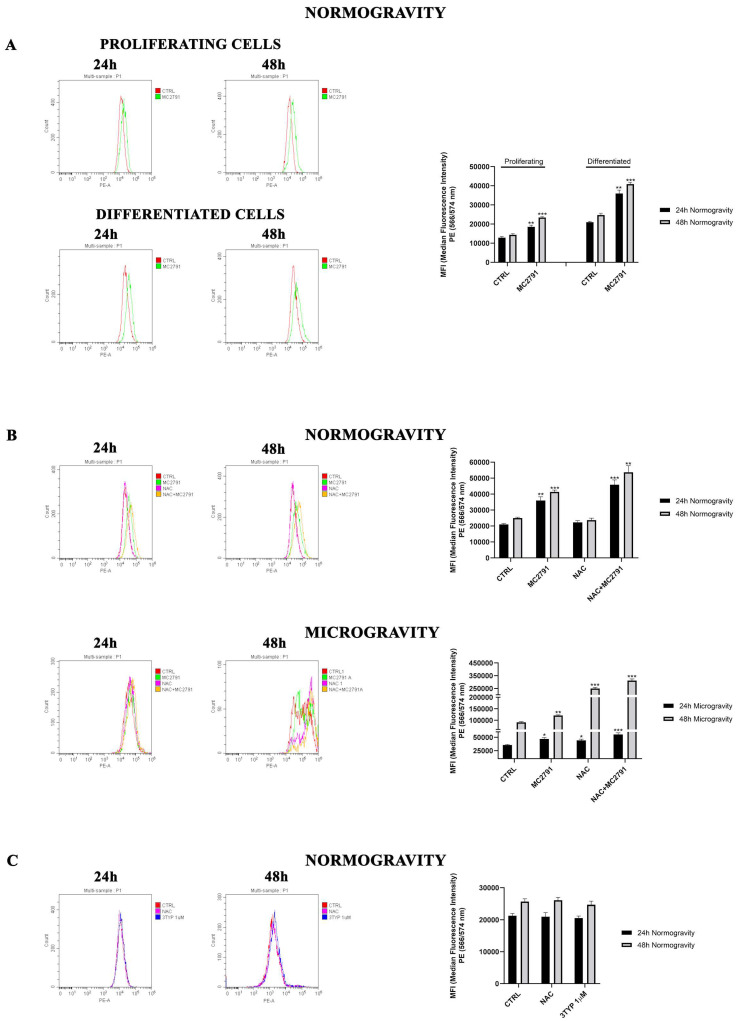
MC2791 increases ROS production in proliferating and differentiated cells. (**A**) ROS production comparison between proliferating and differentiated C2C12, treated or untreated with MC2791 in normogravity conditions. MFI was graphed on the right side of upper panel. (**B**) ROS production in differentiated C2C12 in normogravity and microgravity conditions after 24 and 48 h. MFI was graphed on the right side of middle panel. (**C**) ROS production in differentiated C2C12 in normogravity conditions after 24 and 48 h of treatment with NAC (antioxidant control) and 3-TYP (SIRT3 inhibitor). Median Fluorescence Intensity (MFI) was graphed on the right side of lower panel. Data are representative of three separate experiments. Differences between pairs of groups were analyzed by Student’s *t*-test. * Significantly increased compared with untreated cells. *, *p* < 0.05. **, *p* < 0.01. ***, *p* < 0.001.

**Figure 6 cells-12-00691-f006:**
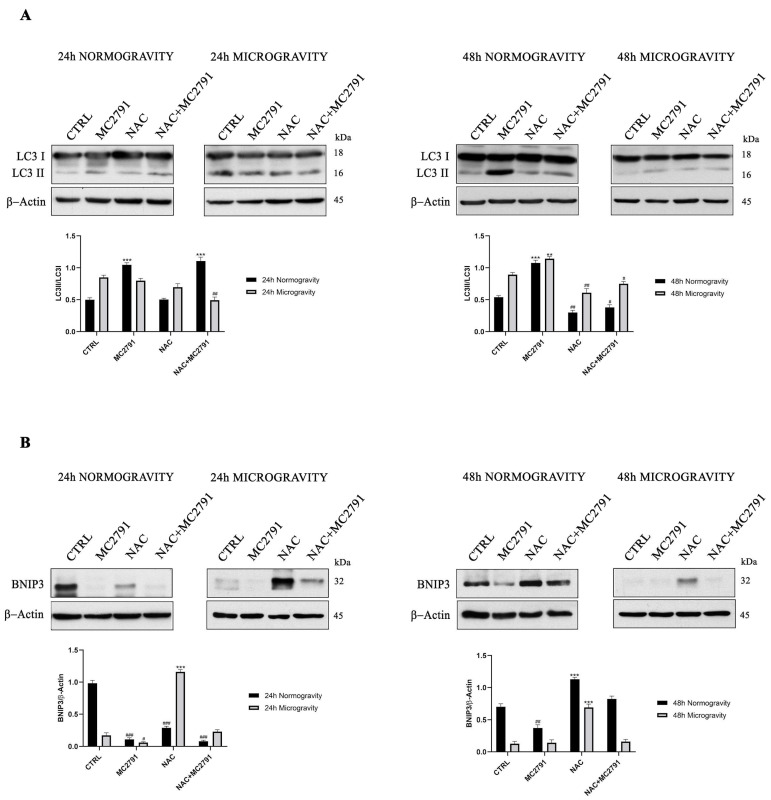
SIRT3 activation enhances autophagy and decreases mitophagy in normogravity and microgravity conditions. C2C12 differentiated cells were treated in normogravity and microgravity conditions as indicated. Expression levels of autophagy marker LC3I and LC3II (**A**) and mitophagy marker BNIP3 (**B**) were determined by Western blot. β-Actin was used as a loading control. Densitometric analysis of the gels was performed as described under Materials and Methods. Data are representative of at least three separate experiments. Differences between pairs of groups were analyzed by Student’s *t*-test. * Significantly increased compared with untreated cells. **, *p* < 0.01. ***, *p* < 0.001. # Significantly decreased compared with untreated cells. #, *p* < 0.05. ##, *p* < 0.01. ###, *p* < 0.001.

**Figure 7 cells-12-00691-f007:**
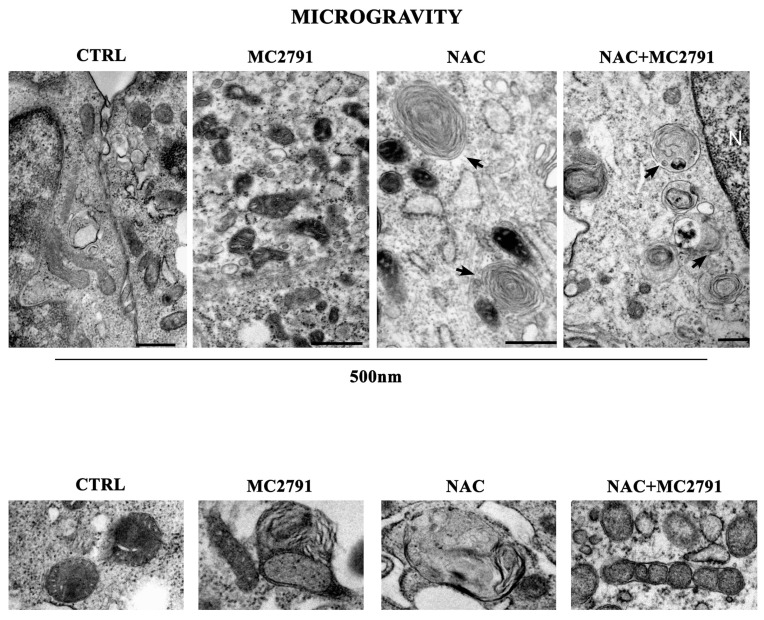
Ultrastructural analysis of differentiated C2C12 under microgravity. C2C12 were differentiated and treated in microgravity conditions. Images show mitochondria morphology and electron density as well as the presence of autophagic structures containing membranes or organelles (black arrows).

## Data Availability

All data generated or analyzed during this study are included in this article.

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
