# Peer review of "Role of SIRT3 in Microgravity Response: A New Player in Muscle Tissue Recovery"

_cells, 2023, doi:10.3390/cells12050691_

Round 1

Reviewer 1 Report

Presented manuscript describes the influence of the treatment with SIRT3 activator on the C2C12 myoblasts cultured under microgravity conditions. Presented manuscript is well-written and seem convincing. The introduction is well planned and cover almost all necessary issues. However, it would be beneficial if this section was supplemented by the information on NAC. From the beginning of the Results section NAC's influence on cells is investigated and the Reader cannot have any clue why. Only in the Section 5.3 it is explained. Moreover, as a non-scpecialist in the issue of oxygen influence on skeletal muscles, mentioning of MnSOD (line 123) does not ring a bell. It should be either ommited, or explained - what is it and what its decreased activity has to do with the C2C12 differentiation process. I have detected some minor errors listed below. On the other hand, the experimental design rise some questions in my opinion.

Section 5.1 - Fig. 1 should be supplemented with presentation of gating strategy (without it one cannot be really sure if sub-G1 cells are indeed cells, not just a debris). Secondly, by looking at the pictures it is not clear what are the populations selected on the histograms (they should be explained). I have also doubts if the ranges of each population on histograms are well selected - for example G2/M population should have twice an amount of DNA as G1 ones.

Section 5.2:

Fig.2 - it is not clear to me how it is possible to do the clonal assay if the cells are differentiated, so post-mitotic. They should not be able to proliferate and to form clones. It is noteworthe that statement "followed by decrease..." in line 339 is misleading. Whole phrase suggests that colonies observed in culture after 24 h disappear afterwards and are not present after another 24 h (i.e. after 48 h of culture), whereas 24 h and 48 h are independent experimental variants as timing corresponds to the time of treatment in normo- vs microgravity and not to the timecourse of culture after seeding.

Fig. 3 in my opinion there should be a scale on the photos. I also think that it would be beneficial if all timepoints were presented at the same magnification (the higher one) - that way morphology of the cells aould be more visible and easier to compare between timepoints. Moreover, is it enough to compare the longitude of cells to claim that they are differentiating on the various rate? In my opinion myotube formation shoud be analysed 2 or 3 days later, for example by counting of the phusion index.

Section 5.3 is well presented and explained, but the selection of factors that are used for the estimation of differentiation progress are controversial in my opinion. Why MyhB/alpha proportion is considered as a determinant of the level of differentiation of muscle cells? Publication that is cited to support this idea is a simple comparison of Myh composition in various fiber types in adult subjects and does not imply analysis of Myh depending on differentiation status [ref 50 in the manuscript].

Secondly, to my knowledge Oct3/4 is a protein characteristic for pluripotent cells and such are not present in regenerating muscle. If such a protein is found in C2C12 cells I would consider it as an artifact. In fact, we are using muslce samples as negative control when trying to detect Oct3/4 in pluripotent stem cells. In my opinion expression of factors specific fp myogenic stem cells should be analysed instead.

Further, manuscript would be better organised if fragment considering ROS analysis was separated as another Section (5.4). On the Fig. 5A I would change the graph so that 24h and 48h constitued bar series (like in Fig. 5B and C) instead of proliferating and differentiating populations. That way comparison of the graphs would be facilitated.

It is interesting why culture in the presence of NAC in microgravity induces the increase in ROS level. I did not find comment on this.

As for general comments - in each figure legend it should be explained which statistical test was used and why.

Minor comments:

line 152 - should be Mus musculus (capital letter not needed)

line 164 lack a dot after "glucose"

line 195 lack of a space between 50 and ug

line 200 extra space in "0.1 %"

line 204 extra space in "Image J"

line 212 lack of the word "antibodies"

line 217 should be into or in instead of "into in"

line 259 I guess it's "100" not "10"

line 263 extra space in "4 degC"

line 286 lack of a come after "microgravity" (in brackets)

line 309 should be "all experiments presented in this figure"

line 595 lack of a dot in Fig. 7

line 615 "mechanis acting" or "mechanism that acts"

line 647 myocytes lack s

line 695 should be "what was demonstrated"

Author Response

Presented manuscript describes the influence of the treatment with SIRT3 activator on the C2C12 myoblasts cultured under microgravity conditions. Presented manuscript is well-written and seem convincing. The introduction is well planned and cover almost all necessary issues. However, it would be beneficial if this section was supplemented by the information on NAC. From the beginning of the Results section NAC's influence on cells is investigated and the Reader cannot have any clue why. Only in the Section 5.3 it is explained. Moreover, as a non-scpecialist in the issue of oxygen influence on skeletal muscles, mentioning of MnSOD (line 123) does not ring a bell. It should be either ommited, or explained - what is it and what its decreased activity has to do with the C2C12 differentiation process. I have detected some minor errors listed below. On the other hand, the experimental design rise some questions in my opinion.

We thank the reviewer for the useful insights that we tried to implement into the manuscript. The new additions have been written in red and highlighted.

We have briefly added information in the Introduction regarding NAC and removed the part regarding MnSOD.

Q: Section 5.1 - Fig. 1 should be supplemented with presentation of gating strategy (without it one cannot be really sure if sub-G1 cells are indeed cells, not just a debris).

A: We thank the reviewer for the advice. We have added the gating strategy as supplementary figures 1 and 2. When selecting the population to analyze we excluded the debris (identified by low size and granulosity), but kept the small-sized cells in the analyzed region to evaluate mortality 

Q: Secondly, by looking at the pictures it is not clear what are the populations selected on the histograms (they should be explained).

A: We added in the legend of Figure 1 the correspondence between the regions selected and the cell cycle phases.

Q: I have also doubts if the ranges of each population on histograms are well selected - for example G2/M population should have twice an amount of DNA as G1 ones.

A: We agree with the reviewer that G2/M population should have twice the amount of G1 population; however, the mean fluorescence intensity of a stained population rarely falls in few channels, it is very often distributed as a bell-shaped gaussian, and the width is dependent on several factors. The cells that we used were differentiated and only a small percentage of the population retained some proliferative activity, and during acquisition the right peak did not fell precisely on 200 (which would have been the expected value, since we center the first peak (the G0/G1) on 100). So, when selecting the limits for population ranges we prefer to set the starting and the ending points of the gates at the points when the curve line changed direction (that is, where a peak become the following one).

Section 5.2:

Q: Fig.2 - it is not clear to me how it is possible to do the clonal assay if the cells are differentiated, so post-mitotic. They should not be able to proliferate and to form clones. It is noteworthe that statement "followed by decrease..." in line 339 is misleading. Whole phrase suggests that colonies observed in culture after 24 h disappear afterwards and are not present after another 24 h (i.e. after 48 h of culture), whereas 24 h and 48 h are independent experimental variants as timing corresponds to the time of treatment in normo- vs microgravity and not to the timecourse of culture after seeding.

A: We apologize for not explaining in a clear way the whole procedure.

The colonies present in the Figure 2 are due to the fact that C2C12 cells are differentiated during the treatment in normo or microgravity, but then are detached and collected in order to be counted and to be plated for colony assay. Moreover, the colony assay is done in proliferating medium (10% FBS) in which cells can proliferate and form colonies. In this way we could better appreciate the differentiation stimulus of MC2791.

We have better explained the whole procedure in the Materials and Methods as well as in the Figure legend and Results section.

Q: Fig. 3 in my opinion there should be a scale on the photos. I also think that it would be beneficial if all timepoints were presented at the same magnification (the higher one) - that way morphology of the cells aould be more visible and easier to compare between timepoints. Moreover, is it enough to compare the longitude of cells to claim that they are differentiating on the various rate? In my opinion myotube formation shoud be analysed 2 or 3 days later, for example by counting of the phusion index.

A: We have replaced images taken at low magnification (10x) so that they are now all taken with a 20x objective. We have inserted bar size in the Figure 3. Since all images were taken at the same magnification, we have inserted the bar at the bottom right of each panel instead of repeating the same bar for every single image.

We have also stained differentiated cells with MHC and Syto (nuclei) in order to measure the fusion index as suggested by the following paper: DOI:10.1186/s13395-021-00284-3 corresponding to reference 44 in the manuscript. We have done it considering only the most significant treatments where differences were clear. Representative images and fusion index measurements are shown in Supplementary Figure 5

Q: Section 5.3 is well presented and explained, but the selection of factors that are used for the estimation of differentiation progress are controversial in my opinion. Why MyhB/alpha proportion is considered as a determinant of the level of differentiation of muscle cells? Publication that is cited to support this idea is a simple comparison of Myh composition in various fiber types in adult subjects and does not imply analysis of Myh depending on differentiation status [ref 50 in the manuscript].

A: We have added new blots in which we measured expression of the differentiation marker myosin heavy chain (MHC) that better recapitulates myoblasts differentiation. However, since the new results agree with the previous ones, we chose to maintain also the MyHβ/MyHα ratio.All the new informations have been added to the Materials and Methods, Figure legend and Results sections.

Q: Secondly, to my knowledge Oct3/4 is a protein characteristic for pluripotent cells and such are not present in regenerating muscle. If such a protein is found in C2C12 cells I would consider it as an artifact. In fact, we are using muslce samples as negative control when trying to detect Oct3/4 in pluripotent stem cells. In my opinion expression of factors specific fp myogenic stem cells should be analysed instead.

A: We have followed the reviewer suggestion and removed the OCT3/4 blots also considering the negative results. However, we have performed new blots for the myogenic stem cells Pax-7 that recapitulates what seen with YAP. The new blots are in Figure 4D and information are added in Materials and Methods, Figure legend, Results and Discussion.

Q: Further, manuscript would be better organised if fragment considering ROS analysis was separated as another Section (5.4). On the Fig. 5A I would change the graph so that 24h and 48h constituted bar series (like in Fig. 5B and C) instead of proliferating and differentiating populations. That way comparison of the graphs would be facilitated.

A: We have added a new section separating the ROS analysis in normogravity and microgravity as suggested. We have also changed the graph in Figure 5A as suggested to homologate it those shown in figure 5B and C.

Q: It is interesting why culture in the presence of NAC in microgravity induces the increase in ROS level. I did not find comment on this.

A: We commented on the result showing an increase of ROS in the presence of NAC in the Discussion (lines 636-641). Briefly, we believe that although more experiments are required to better understand such a result, the increase is due to the presence of microgravity that, somehow, is increasing the pro-oxidant effect of NAC, a situation similar to the prooxidant effect reported for NAC when used at high concentrations.

Q: As for general comments - in each figure legend it should be explained which statistical test was used and why.

A: We have added the statistical test adopted in the figure legends

Minor comments:

line 152 - should be Mus musculus (capital letter not needed)

line 164 lack a dot after "glucose"

line 195 lack of a space between 50 and ug

line 200 extra space in "0.1 %"

line 204 extra space in "Image J"

line 212 lack of the word "antibodies"

line 217 should be into or in instead of "into in"

line 259 I guess it's "100" not "10"

line 263 extra space in "4 degC"

line 286 lack of a come after "microgravity" (in brackets)

line 309 should be "all experiments presented in this figure"

line 595 lack of a dot in Fig. 7

line 615 "mechanis acting" or "mechanism that acts"

line 647 myocytes lack s

line 695 should be "what was demonstrated"

A: We have corrected all the minor mistakes and typos.

Reviewer 2 Report

The authors demonstrated that the SIRT3 activation could have beneficial effects on microgravity-induced muscle tissue damage. In general, the manuscript is well-written, and the methods and results are convincing. The discussion is balanced by discussing the present results in the context of the results of the existing literature. In general, the findings of the authors reflect the conclusions of the manuscript. Using in vitro cellular models, the authors confirm the harmful effects of microgravity already described in several publications using different cellular models. Indeed, microgravity represents a stress factor and the findings are not surprising. However, the authors present some novel findings demonstrating that activation of the SIRT3 pathway may have some beneficial effects against the harmful impact of microgravity on muscle cells. In summary, the manuscript should be written concisely and focuses on the main findings of the manuscript.

I have some suggestions that may help to improve the manuscript. To my opinion, the length and the repetitions of the manuscript make the reading of the manuscript very tedious. The whole manuscript should be more concise; otherwise, the reader loses enthusiasm for reading the manuscript. The introduction is two pages giving too many details about the effects of microgravity on some other cellular systems; for instance, on page 2, lines 59-75 can be summarised in 1 to 2 sentences. To my opinion, the introduction can be reduced to one page focusing on the key effects of the microgravity to muscle cells.

The results part can be also significantly reduced and made this part more concise e.g., lines 478 to 487; this paragraph can be reduced to one sentence and should be included in the introduction or discussion part. Several parts in the results part should be shortened and moved to the Introduction or Discussion part.

Whereas authors cited many references that are not urgently relevant in the context of their findings, other much more related findings are not cited; Here 2 examples. from PubMed:

Microgravity-induced stress mechanisms in human stem cell-derived cardiomyocytes.

Acharya A, Nemade H, Papadopoulos S, Hescheler J, Neumaier F, Schneider T, Rajendra Prasad K, Khan K, Hemmersbach R, Gusmao EG, Mizi A, Papantonis A, Sachinidis A.iScience. 2022 Jun 11;25(7):104577. doi: 10.1016/j.isci.2022.104577. eCollection 2022 Jul 15.

Simulated Microgravity Modulates Differentiation Processes of Embryonic Stem Cells.

Shinde V, Brungs S, Henry M, Wegener L, Nemade H, Rotshteyn T, Acharya A, Baumstark-Khan C, Hellweg CE, Hescheler J, Hemmersbach R, Sachinidis A.Cell Physiol Biochem. 2016;38(4):1483-99. doi: 10.1159/000443090. Epub 2016 Apr 4.PMID: 27035921 Free article.

These references should be cited.

There are some small errors in the text that can be improved by a more careful reading of the manuscript; for instance line 189 (mmol/L); line 184 mMol/L. The units should be unique.  

Author Response

The authors demonstrated that the SIRT3 activation could have beneficial effects on microgravity-induced muscle tissue damage. In general, the manuscript is well-written, and the methods and results are convincing. The discussion is balanced by discussing the present results in the context of the results of the existing literature. In general, the findings of the authors reflect the conclusions of the manuscript. Using in vitro cellular models, the authors confirm the harmful effects of microgravity already described in several publications using different cellular models. Indeed, microgravity represents a stress factor and the findings are not surprising. However, the authors present some novel findings demonstrating that activation of the SIRT3 pathway may have some beneficial effects against the harmful impact of microgravity on muscle cells. In summary, the manuscript should be written concisely and focuses on the main findings of the manuscript.

A: We thank the reviewer for the careful reading of the manuscript and for the useful suggestions. We have modified the manuscript accordingly. All changes have been written in red and highlighted.

Q: I have some suggestions that may help to improve the manuscript. To my opinion, the length and the repetitions of the manuscript make the reading of the manuscript very tedious. The whole manuscript should be more concise; otherwise, the reader loses enthusiasm for reading the manuscript. The introduction is two pages giving too many details about the effects of microgravity on some other cellular systems; for instance, on page 2, lines 59-75 can be summarised in 1 to 2 sentences. To my opinion, the introduction can be reduced to one page focusing on the key effects of the microgravity to muscle cells.

A: We have shortened the Introduction that is now little bit over 1 page focusing mostly on muscle tissue and microgravity.

Q: The results part can be also significantly reduced and made this part more concise e.g., lines 478 to 487; this paragraph can be reduced to one sentence and should be included in the introduction or discussion part. Several parts in the results part should be shortened and moved to the Introduction or Discussion part.

A: We have revised and shortened the Results and Discussion sections trying to point out the most significant results and findings.

Q: Whereas authors cited many references that are not urgently relevant in the context of their findings, other much more related findings are not cited; Here 2 examples. from PubMed:

Microgravity-induced stress mechanisms in human stem cell-derived cardiomyocytes.

Acharya A, Nemade H, Papadopoulos S, Hescheler J, Neumaier F, Schneider T, Rajendra Prasad K, Khan K, Hemmersbach R, Gusmao EG, Mizi A, Papantonis A, Sachinidis A.iScience. 2022 Jun 11;25(7):104577. doi: 10.1016/j.isci.2022.104577. eCollection 2022 Jul 15.

Simulated Microgravity Modulates Differentiation Processes of Embryonic Stem Cells.

Shinde V, Brungs S, Henry M, Wegener L, Nemade H, Rotshteyn T, Acharya A, Baumstark-Khan C, Hellweg CE, Hescheler J, Hemmersbach R, Sachinidis A.Cell Physiol Biochem. 2016;38(4):1483-99. doi: 10.1159/000443090. Epub 2016 Apr 4.PMID: 27035921 Free article.

These references should be cited.

 A: After revising the whole manuscript, we have removed some references and inserted those suggested by the reviewer. 

Q: There are some small errors in the text that can be improved by a more careful reading of the manuscript; for instance line 189 (mmol/L); line 184 mMol/L. The units should be unique.  

A: We have corrected all the minor mistakes and typos.

Round 2

Reviewer 1 Report

The presented manuscript gained a lot by modifications that were made. There are still some minor typos, but altogether I find the manuscript convincing. The main issue I would like to point to is the style of figure description and results description whe reffering to figures. What I mean is that the Authors write in a way that suggests the figures were part of the experiments not simply their description. To cite some examples: "in the panel cell were...", "in figure we analyzed...", "following results of figure...". In my opinion all such sentences should be modified in a way that states the fact, i.e. e.g. "figure presents...". Manuscript should also be checked for the time accordance (for example: in line 463 should be "analyzed" or "performed" in line 555). Some detailed issues:

line 22 should be Earth instead of earth

line 30 should be "... were measured..."

abstract goes into too much details in lines 32-35

line 43 should be space instead of Space

line 299/300 should be "as described" instead of "as previously described"

line 300/301 gates should be described with majuscule (P3, P4, etc)

line 321 post mitotic should be written together

line 369 should be "it is clear that..." not "it is clear how"

line 372/373 should be "Therefore, we aimed"

line 475 should be "NAC treatment"

Author Response

Q:

The presented manuscript gained a lot by modifications that were made. There are still some minor typos, but altogether I find the manuscript convincing. The main issue I would like to point to is the style of figure description and results description whe reffering to figures. What I mean is that the Authors write in a way that suggests the figures were part of the experiments not simply their description. To cite some examples: "in the panel cell were...", "in figure we analyzed...", "following results of figure...". In my opinion all such sentences should be modified in a way that states the fact, i.e. e.g. "figure presents...". Manuscript should also be checked for the time accordance (for example: in line 463 should be "analyzed" or "performed" in line 555). Some detailed issues:

line 22 should be Earth instead of earth

line 30 should be "... were measured..."

abstract goes into too much details in lines 32-35

line 43 should be space instead of Space

line 299/300 should be "as described" instead of "as previously described"

line 300/301 gates should be described with majuscule (P3, P4, etc)

line 321 post mitotic should be written together

line 369 should be "it is clear that..." not "it is clear how"

line 372/373 should be "Therefore, we aimed"

line 475 should be "NAC treatment"

A:

We thank the reviewer for the useful suggestions.

We have changed the figures legends and results description to better describe the figures.

We have also checked the manuscript for time accordance.

Finally, we have corrected the typos and removed the sentence in the abstract (lines 32-35) with too many experimental details.

Reviewer 2 Report

Authors responded satisfactorily to my points of criticism and appropriate changes were made 

Author Response

We thank the reviewer for helping us to improve the manuscript.